

# *Loancorhynchus catrillancai* gen. et sp. nov., a new swordfish (Xiphioidei, Blochiidae) from the Middle Eocene of central Chile

Rodrigo A. Otero

Laboratorio de Ontogenia y Filogenia, Departamento de Biología, Facultad de Ciencias, Universidad de Chile, Santiago, Chile
Consultora Paleosuchus Ltda, Santiago, Chile

## ABSTRACT

This contribution describes the skull remains of a swordfish (Perciformes, Xiphioidei), recovered from Middle Eocene beds of central Chile. Comparison with known fossil and extant xiphioids reveals derived traits only present in the Neogene swordfish *Xiphias gladius* (Xiphiidae, Xiphiinae), these being a elongated rostrum composed of pre-maxillaries and possible prenasals, a dorsoventrally high and slender hyomandibular-metapterygoid complex, and a rounded, convex operculum. Also, strong ridges and sulci are present in the dorsal part of the rostrum, a feature only present in the billfish genera *Aglyptorhynchus* and *Xiphiorhynchus*, and in the swordfish genus *Blochius*. In addition, the specimen also has villiform teeth in the rostrum and lower jaw, a feature previously documented only in the Paleocene genus *Hemingwaya*. Such a unique combination of characters in the specimen allows classifying it as a new genus and species, *Loancorhynchus catrillancai*. Phylogenetic analyses obtained *Xiphiorhynchus* outside the Xiphiidae, suggesting instead narrow relationships to the Paleocene-Eocene genera *Hemingwaya* + *Palaeorhynchus* + *Homorhynchus*. *Loancorhynchus* is obtained as an intermediate form between *Xiphias* and *Blochius*. The specimen represents the first Paleogene swordfish described in the southeastern Pacific.

Corresponding author
Rodrigo A. Otero,
rotero@paleosuchus.cl,
otero2112@gmail.com

## INTRODUCTION

The Xiphioidei (Osteichthyes, Perciformes) is a group of fishes considered to be monophyletic, being characterized by elongated premaxillaries forming a rigid rostrum, and by having villiform teeth (*Fierstine, 2006*), although the latter can be secondarily lost during the ontogeny of some taxa (e.g., *Xiphias gladius*). To date, five family-level groups are known: Hemingwayidae, Palaeorhynchidae, Blochiidae, Xiphiidae and Istiophoridae (*Fierstine, 2006*; *Gottfried, Fordyce & Rust, 2013*). The two latter clades, Xiphiidae and Istiophoridae, are the only lineages with extant representatives.

Late Cretaceous–Paleogene records of xiphioids are scarce in the Weddellian Province. *Zinsmeister (1979)* defined it as a large province including southern South America,

Antarctica, New Zealand and part of Australia, isolated between the Late Cretaceous–Paleogene by the geography of Gondwana landmass and prevailing oceanic circulation, resulting in the development of highly endemic marine faunas (*Zinsmeister, 1982*). The first mention of a Paleogene Xiphioidei from the Weddellian Province concerns a large, isolated vertebra from lower Eocene levels of La Meseta Formation, in Marambio (=Seymour) Island, Antarctica, referred to cf. *Xiphiorhynchus* (*Cione, Reguero & Elliot, 2001*). *Friedman & Otero (2009)* added to this record the preliminary description of a fragmentary rostrum and dentaries referred to as Xiphiorhynchine indet., which was recovered from Paleogene levels of central Chile. This material is part of the same specimen described here. After that, *Gottfried, Fordyce & Rust (2013)* described a new species of palaeorhynchid, *Aglyptorynchus hakataramea*, from late Oligocene levels exposed in South Canterbury, South Island, New Zealand. A fourth specimen (GS 13924; GNS Science, Lower Hutt, New Zealand), still awaiting description, belongs to an articulated skull embedded in a concretion, recovered from Eocene levels of Hampden Beach, Otago, South Island, New Zealand (*Campbell et al., 2013*). Locally, Neogene records of xiphiids and istiophorids are well-known in the Miocene-Pliocene of Chile (*Long, 1993*; *Walsh, 2001*; *Pyenson et al., 2014*). A mention of cf. *Xiphiorhynchus* from the Miocene-Pliocene of Peru was provided by *De Muizon & DeVries (1985)*, but the material was not figured neither a repository was provided.

This contribution describes the only xiphioid recovered to date from Paleogene beds of the southeastern Pacific. After further preparation, new skull elements are added to its first preliminary description by *Friedman & Otero (2009)*. In addition, the stratigraphic provenance of the specimen is clarified here. As a result, a new genus and species are here proposed. The fossil is found to be Middle Eocene in age, representing the first significantly diagnostic specimen from the Paleogene of South America, and adding to the knowledge of the sparse Paleogene xiphioids known along the Southern Hemisphere.

## LOCALITY AND GEOLOGIC SETTING

Loanco is a small cove placed 350 km south from Santiago (Fig. 1A). In this locality, rich fossiliferous marine rocks crop out along the coastal cliffs and the intertidal surface. Beds exposed north of Loanco have been recognized as equivalent to the Late Cretaceous Quiriquina Formation (*Tavera, 1987*), assigned to the upper Maastrichtian based on biostratigraphy and stratigraphic correlations (*Otero, 2015*; *Castro et al., 2016*). A second sedimentary and also marine unit was recently found cropping out south of Loanco. From the latter, *Suárez & Otero (2008)* recovered associated teeth, vertebrae and mandibular cartilague belonging to a single individual referred to the genus *Isurolamna* (Chondrichthyes, Elasmobranchii). Previous geologic studies have referred the outcrops in south of Loanco as Late Cretaceous in age (*Sernageomin, 2003*). During 2012, new fieldwork in this locality has provided material referable to the nautiloid genus *Aturia* (*Zambrano, Nielsen & Stinensbeck, 2014*) and *Imaizula araucana* (recovered during fieldwork of this research), which are unequivocally Cenozoic taxa. More recently, *Otero (2015)* identified the presence of chondrichthyan teeth referable to *Macrorhizodus praecursor* (SGO.PV.6633; SGO, Paleontología de Vertebrados, Museo Nacional de Historia Natural, Santiago, Chile).

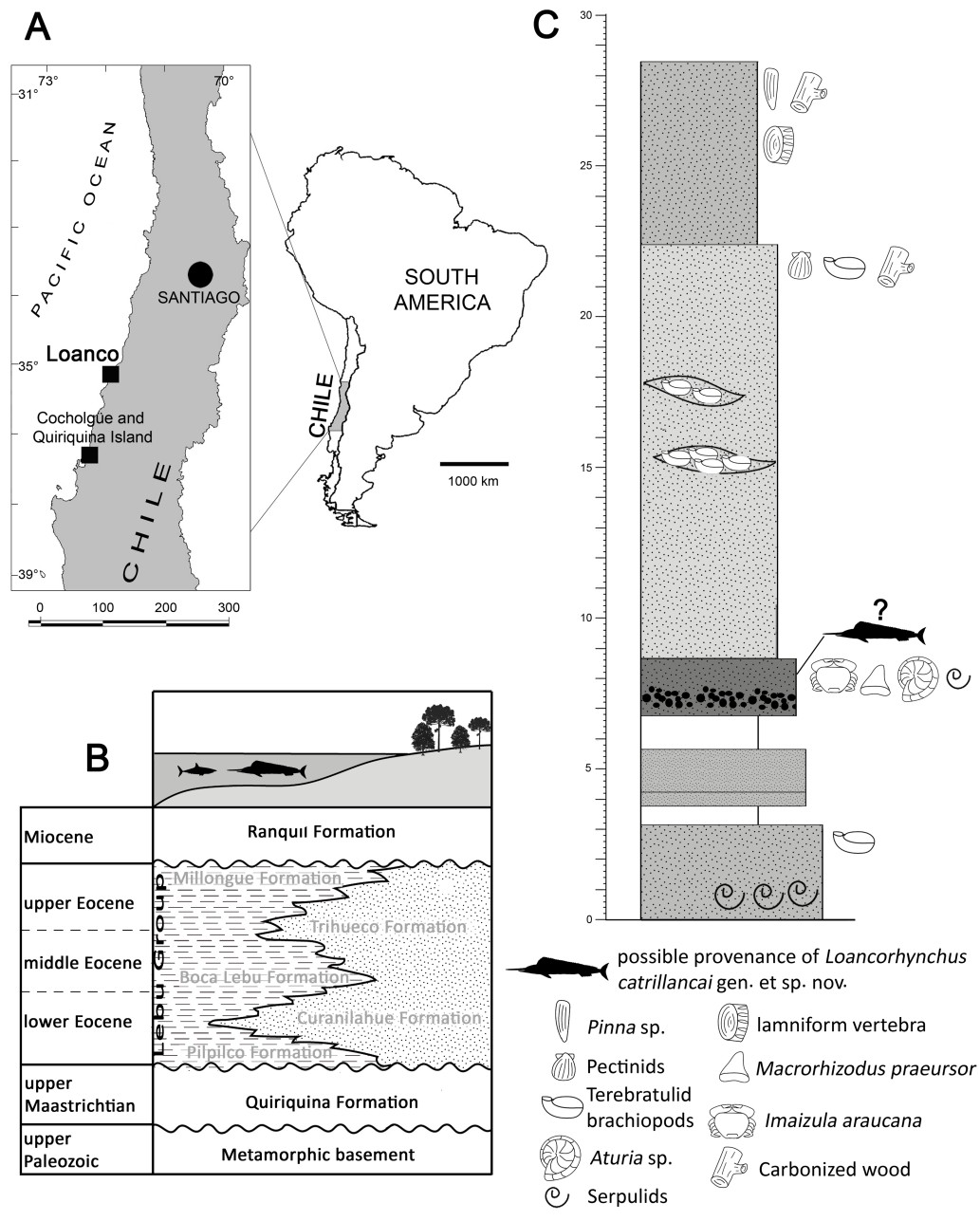

**Figure 1** **Locality and stratigraphy of SGO.PV.6634.** (A) Map indicating the locality of Loanco, where SGO.PV.6634 was recovered. (B) Scheme of the transgressive-regressive deposits in the Arauco Basin, indicating the respective units. Modified from *Charrier, Pinto & Rodríguez, 2007*. (C) Stratigraphic column of the best Paleogene outcrop currently available (2016) in the coast south of Loanco. The unit is larger, however, this is covered by recent sands and by tsunami deposits after February 27, 2010. Sediment attached to SGO.PV.6634 is coincident with the fossil-bearing level containing *Aturia* sp. and *Imaizula araucana*.

The combined biochron of this assemblage indicates a Middle Eocene age for the fossil-bearing levels.

The Quiriquina Formation and the Paleogene marine units of central Chile were deposited in a continental shelf which is part of a single realm known as the Arauco Basin (*Mordojovic, 1981*), extended at least between 33° and 37°S. In this context, Paleogene units have been recorded along the whole basin, being deposited in angular unconformity over Late Cretaceous beds of the Quiriquina Formation or equivalent units. In Cocholgüe and the Quiriquina Island, 120 km south from Loanco, the Quiriquina Formation is unconformably overlain by Paleogene sediments of the Concepción Group (*sensu Charrier, Pinto & Rodríguez, 2007*; =Lebu Group in *Cecioni, 1968*; *Le Roux, Nielsen & Henríquez, 2008*). Several Paleogene units have been characterized in the Arauco Basin, representing different stages of a transgressive-regressive cycle. After a Paleocene hiatus, from base to roof, a first shallow transgression is represented by the marine-to-continental, upper Paleogene-lower Eocene Pilpilco Formation (*Muñoz Cristi, 1968*), conformed by ca. 150 m of fine-grained, greenish sandstones with clay intercalations. The unit unconformably overlies the metamorphic basement and the Quiriquina Formation. The Pilpilco Formation is covered by a conformably deposited, regressive unit represented by the continental, coal-rich sandstones of the Curanilahue Formation (*Tavera, 1942*; *Martínez-Pardo, 1968a*) assigned to the lower Eocene. Over the latter and in conformable contact lays the Boca Lebu Formation (*Muñoz Cristi, 1946*; *Muñoz Cristi, 1968*), which comprises a variable section between 250 and 600 m of marine, fossiliferous green sandstones with glauconite. The upper regression is represented by the continental Trihueco Formation (*Muñoz Cristi, 1946*; *Muñoz Cristi, 1968*) comprised by 3–10 m thick banks with blue-to-grey sandstones with intercalations of brown, coal-rich shales and coal seams. The final transgression is represented by the Millongue Formation (*Muñoz Cristi, 1946*). This unit was characterized by *García (1968)* as 270 m of thin beds of shales and limestones with leaf prints and frequent fragments of carbonized wood, representing a marine-continental environment. Following this geologic scheme (Fig. 1B), the fossil-bearing section studied here could be referred to one of these marine units (i.e., basal Pilpilco Formation; Boca Lebu Formation or Millongue Formation).

The studied section (Fig. 1C) is comprised from base to roof, by: 3 m of greenish sandstones with banks of serpulids; 0.5 m with no outcrop; 2 m of reddish sandstones; 2 m with no outcrop; 1.8 m of mid-grained sandstone, brown, with micaceous fragments, and having a basal fine conglomerate. Fossils include teeth of *Macrorhizodus praecursor*, a phragmocone of the nautiloid *Aturia* sp. (SGO.PI.6776), and an articulated individual of *Imaizula araucana*; 14.3 m of mid-grained, yellow-to-green sandstone with lenses including banks of terebratulid brachiopods. Isolated terebratulids, pectinids and carbonized wood fragments occur near the roof of the level; 6 m of mid-grained, reddish to brown sandstones, including few individuals of the bivalvian *Pinna* sp. in life position, as well as an isolated lamniform vertebra, serpulids, carbonized wood fragments, and *Teredolites* isp. Lithologically, these strata could be referred to the Boca Lebu or Millongue formations; however, the presence of *Pinna* sp. has been previously considered as a proxy for distinguishing both units, because the occurrence of this genus seems to be restricted

only to the Millongue Formation (*Martínez-Pardo, 1968b*: p. 100). Adding to these facts, a Middle Eocene age is proposed based on the combined biochron of the faunal assemblage. Then, the studied unit is here proposed to be equivalent to the Millongue Formation. Several outcrops of the same unit in south Loanco were documented by the author during the extraction of SGO.PV.6635 (associated chondrichthyan remains referred to *Isurolamna* sp.); however, these were later covered by sand banks after the large earthquake and subsequent tsunami of central Chile in February 27, 2010. Currently, the unit is only visible during low tides along the coastal line south from Loanco, but its extension under the recent sands reaches hundreds of meters. The sediment attached to SGO.PV.6634 is remarkably similar to the fossil-bearing level including *Aturia* sp. and *Imaizula araucana*; however, the lack of historical information regarding its collection (see text below) makes impossible to assure a precise stratigraphic provenance.

## MATERIAL AND METHODS

**Historical background of the studied specimen**—SGO.PV.6634 (Fig. 2) was recovered by H Fuenzalida, former director of the Museo Nacional de Historia Natural (MNHN, National Museum of Natural History, Santiago, Chile), during 1935 from rocks exposed south of Loanco, as is accounted by his personal field notes archived in the aforementioned institution. The collected material also included ammonoids, bivalvians and plesiosaur remains; however, these were recovered without stratigraphic data. Later, the collection of Fuenzalida was studied by *San Martín (1946)*, who assigned a 'Senonian' age to the assemblage based on ammonoids, and recognizing affinities with the fauna of the Quiriquina Formation. Among the determinations by *San Martín (1946)*, the plesiosaur remains were considered as Cenozoic remains of indeterminate cetaceans. The swordfish remains were not prepared at that moment, being interpreted as *Teredo* remains. Since then, the whole collection was housed in the MNHN. Probably because of its antiquity, the collection was not included in any catalog, being re-discovered during 2008 by S. Soto-Acuña (Universidad de Chile). After this, a first inspection of the 'wood remains' revealed that they are actually bone remains. Subsequently, part of the material was mechanically prepared by the author during 2008 and 2009. The rostrum and dentaries were preliminary presented and identified as an indeterminate xiphiorhynchine by *Friedman & Otero (2009)*. These authors also gave to the specimen, for the first time, a formal numeration and repository (SGO.PV.6634). Further preparation was carried out by the author during 2012 and 2013, uncovering new elements of the skull. Anatomical identifications were made during April, 2013 by direct comparison with the holotype of *Aglyptorhynchus hakataramea Gottfried, Fordyce & Rust, 2013*, housed in the Department of Geology of the Otago University, Dunedin, New Zealand. Further anatomical identifications follow the classical work of *Gregory & Conrad (1937)*. Considering the osteological nomenclature of billfishes in general, there is a controversial issue regarding the presence of prenasal bone (*Schultz, 1987*; *Fierstine, 1990*; *Fierstine & Voigt, 1996*). Resolving this issue is out of the scope of this contribution. However, this research follows the uncertain denomination of the dorsomedial rostrum element in question as prenasals, as was done by *Fierstine (1990)*. The same element was considered by *Gregory & Conrad (1937)* as the nasal.

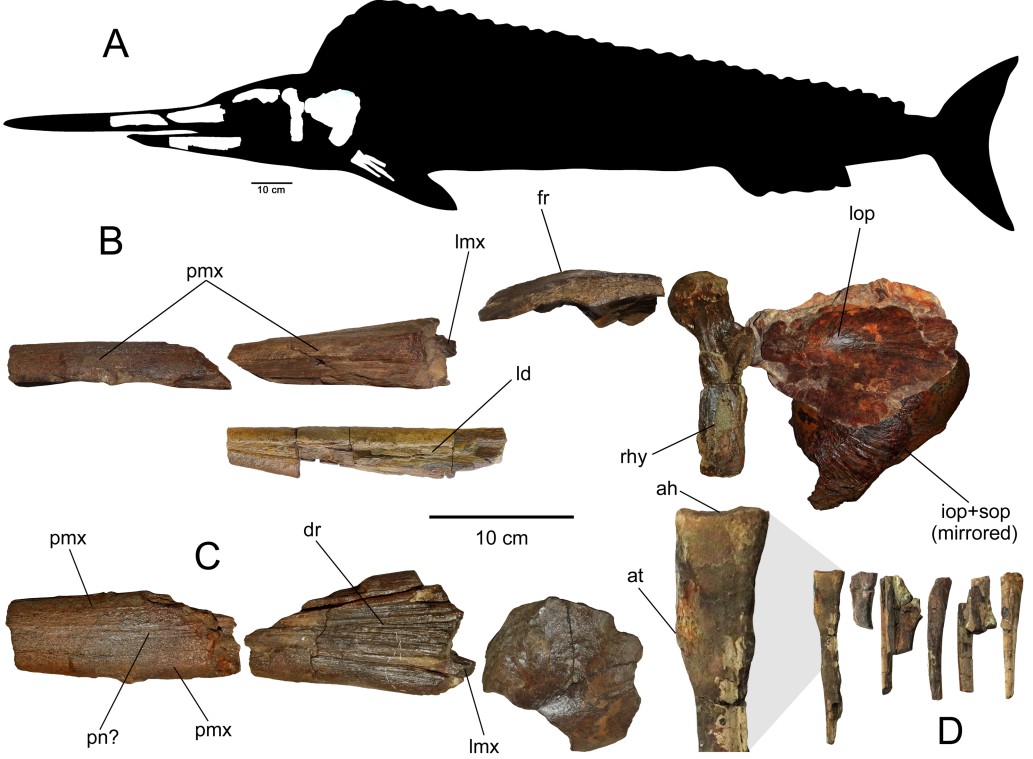

**Figure 2** **Skull elements of SGO.PV.6634, holotype of *Loancorhynchus catrillancai* gen. et sp. nov.** (A) Estimated body outline (based on *Fierstine & Monsch, 2002*). (B) Detail of the available skull elements of SGO.PV.6634. (C) Dorsal view of the rostral elements (D) Fin rays, with detail of the proximal part of the first right ray. Anatomical abbreviations: ah, articular head; at, anterior tuberosity; dr, dorsal ridges; fr, frontal; ld, left dentary; lmx, left maxillary; lop, left operculum; pmx, premaxillary; pn?, prenasals?; rhy, right hyomandibular.

**Phylogenetic analysis**—Phylogenetic analysis used a modified version of the datamatrix with twelve taxa (fossil and extant) and 25 characters provided by *Fierstine & Monsch (2002)*. Modifications here introduced include three new characters (26–28) and addition of new states for characters 4, 11 and 15 (see Data S1). Original dataset of Istiophoridae was replaced with the scorings of *Istiophorus albicans* and *Makaira nigricans* (taken from *Nakamura, 1985*). Original dataset of *Aglyptorhynchus* was also replaced with the scores of *Aglyptorhynchus hatakaramea*, which is a fairly complete specimen from the Eocene of New Zealand (*Gottfried, Fordyce & Rust, 2013*). Also, the scores of *Hemingwaya sarissa Sytchevskaya & Prokofiev, 2002*; and SGO.PV.6634 were added to the datamatrix. Few taxa in the original datamatrix of *Fierstine & Monsch (2002)* were pruned based in different criteria:

*Cylindracanthus*—This genus lacks a subdermal tooth base attachment (*Grandstaff et al., 2017*), leading to the conclusion that *Cylindracanthus* does not belong to any billfish (=Xiphioidei), which indeed have subdermal tooth base attachments. Previously, *Parris & Grandstaff (2001)* suggested that *Cylindracanthus* could be related to Acipenseriformes.

*Hemirhabdorhynchus*—*Fierstine & Monsch (2002)* originally excluded this genus from phylogenetic analysis based in its ambiguity. This observation is here followed. Even more, no further clarification regarding the anatomy and taxonomy of *Hemirhabdorhynchus* was provided since *Schultz (1987)*. *Fierstine & Monsch (2002)* mentioned *Schultz (1987)* as the source of information used by them for scoring of this genus.

*'Blochius' moorheadi*—This was detected as an unstable taxon with the IterPcr Script (*Pol & Escapa, 2009*). *Fierstine & Monsch (2002)* also found this taxon as unstable, pruning it from the returned reduced consensus tree (*Fierstine & Monsch, 2002*: fig. 7).

*Pseudotetrapturus luteus*—This taxon was removed for reducing ambiguity. Besides the rostrum, the skull of the holotype is naturally crushed, and the postcranial skeleton is currently lost (*Monsch & Bannikov, 2011*).

The modified datamatrix includes 11 taxa and 28 characters. This was analyzed with TNT software (*Goloboff, Farris & Nixon, 2008*).

**Permissions**—The studied fossil was collected in 1935. At that time, field permissions for paleontologic excavation in Chile were not required. Futher fieldwork in 2012 was granted by the Chilean authority in charge, Consejo de Monumentos Nacionales, in document Ord. CMN No 2962/2009 and currently, in Ord. CMN No 3793/2014.

**Nomenclatural acts**—New names contained in the electronic version of this article are effectively published under the International Commission on Zoological Nomenclature (ICZN) from the electronic edition alone. This published work and the nomenclatural acts it contains have been registered in ZooBank, the online registration system for the ICZN. The ZooBank LSIDs (Life Science Identifiers) can be resolved and the associated information viewed through any standard web browser by appending the LSID to the prefix "http://zoobank.org/". The LSID for this publication are:

Genus name: LSID:urn:lsid:zoobank.org:act:FA566753-31D4-45B5-9E7A-03F8C7526E7D

Species name: LSID: urn:lsid:zoobank.org:act:8FFA9EE9-A81B-4468-9FBA-9D6C306D86C8

Publication: LSID:urn:lsid:zoobank.org:pub:ED22325A-7C78-469A-9985-365C31064652

The electronic version of this article in Portable Document Format (PDF) will represent a published work according to the International Commission on Zoological Nomenclature (ICZN), and hence the new names contained in the electronic version are effectively published under that Code from the electronic edition alone. This published work and the nomenclatural acts it contains have been registered in ZooBank, the online registration system for the ICZN. The ZooBank LSIDs (Life Science Identifiers) can be resolved and the associated information viewed through any standard web browser by appending the LSID to the prefix http://zoobank.org/. The LSID for this publication is: LSID:urn:lsid:zoobank.org:pub:ED22325A-7C78-469A-9985-365C31064652.

The online version of this work is archived and available from the following digital repositories: PeerJ, PubMed Central and CLOCKSS.

# SYSTEMATIC PALEONTOLOGY

Order PERCIFORMES *sensu Johnson & Patterson, 1993*
Suborder XIPHIOIDEI *sensu Gosline, 1968*
Family BLOCHIIDAE (*Bleeker, 1859*)
*Genus LOANCORHYNCHUS* nov.

**Type Species**—*Loancorhynchus catrillancai* gen. et sp. nov., by monotypy.

**Derivation of Name**—Genus name after its type locality, Loanco, Región del Maule, central Chile.

**Diagnosis**—As for the single known species, below.

*LOANCORHYNCHUS CATRILLANCAI* gen. et sp. nov.
(Figs. 2–6)
*Teredo* sp.: *San Martín, 1946*: p. 46.
Xiphiorhynchinae indet.: In *Friedman & Otero, 2009*.

**Holotype**—SGO.PV.6634, a dissarticulated three-dimensional skull preserving most of the rostrum (premaxillaries and prenasals?), fragments of both dentaries, a fragment of the left maxilla, left frontal, left metapterygoid+hyomandibular, both operculi, and several branchystegal bones.

**Derivation of Name**—Honoring the memory of the Mapuche leader Camilo Catrillanca.

**Locality, Horizon and Age**—Loanco, Región del Maule. Indeterminate levels of the Millongue Formation cropping out south Loanco. Middle Eocene.

**Diagnosis**—Rostrum much longer than the dentaries, with villiform teeth reaching the posterior part of the rostrum; rostrum and dentaries with villiform teeth over their occlusal and latero-occlusal surface; rostrum with clearly oval, dorsally compressed cross-section, intermediate between *Xiphias* and *Xiphiorhynchus*; operculum rounder and broader than in *Blochius longirostris* but dorsoventrally more compressed than in *Xiphias gladius*; hyomandibular/metaethmoid complex more gracile than in *Blochius* but broader than in *Xiphias*.

**Other distinguishing characters**—*Loancorhynchus catrillancai* skull has premaxillaries joined into a pointed rostrum and a rostrum usually longer than the lower jaw. Both features have been considered as diagnostic of the family Blochiidae by *Fierstine & Monsch (2002)*. Unique combination of characters include features found in Hemingwayidae (*Sytchevskaya & Prokofiev, 2002*), Xiphiorhynchinae (*Regan, 1909*), Xiphiinae (sensu *Fierstine, 2006*) and Blochiidae (*Bleeker, 1859*) *Loancorhynchus catrillancai* possesses villiform teeth along the premaxillaries and also in the occusal surface of the dentaries. Such feature is only documented in the *Hemingwaya sarissa Sytchevskaya & Prokofiev, 2002*. *Loancorhynchus catrillancai* also possesses a thick, dorsoventrally compressed rostrum formed mostly by premaxillaries extended posteriorly on each side, while its dorsal midline is conformed by an anterior extension of the prenasals(?), and posteriorly, by a short extension of

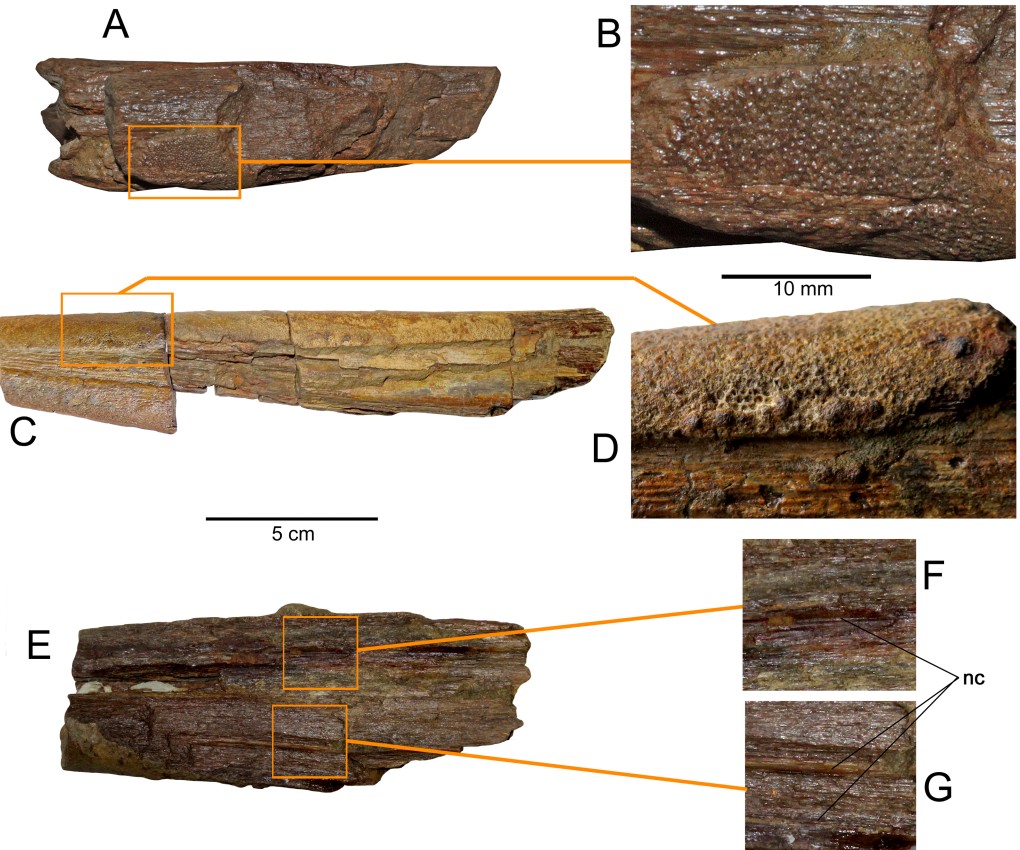

**Figure 3** *Loancorhynchus catrillancai* **gen. et sp. nov. SGO.PV.6634, holotype. Distribution of villi-form teeth and canali.** (A) posterior part of the rostrum in lateral view. (B) detail of the villiform teeth present in the posterior part of the right maxillary. (C) Left dentary in left lateral view. (D) detail of the villiform teeth in the anterior part of the left dentary. (E) Ventral view of the anterior rostrum fragment. (F, G) Detail of the internal paired canali in the rostrum.

the dermethmoid, as it occurs in *Xiphias*; It also shares with *Xiphias* the presence of a metapterygoid fused to the hymandibular, forming a dorsoventrally high complex with a prominent dorsal articulation, and a subrounded, laterally convex operculum (*Gregory & Conrad, 1937*). Traits shared with Xiphiorhynchinae include a lower jaw shorter than the rostrum but comparatively larger than that of *Xiphias*; posterior part of the rostrum with strong folds and sulci, as those present in *Aglyptorhynchus*, *Xiphiorhynchus* and *Blochius*; additional traits include a lower jaw with dentaries contacted in a large craniocaudal symphysis as that present in *Pseudotetrapturus*, *Blochius* and all xiphiids.

**Taxonomic placement of *Loancorhynchus catrillancai*—**The phylogenetic analysis (see 'Discussion' below) confirms the placement of Blochiidae as the sister taxon of *Xiphias gladius* (i.e., Xiphiidae), as previously obtained by *Fierstine & Monsch (2002)* and by *Fierstine (2006)*. *Loancorhynchus catrillancai* has a set of derived traits shared with *Xiphias*, including a rostrum formed by premaxillaries+prenasals(?), a dorsoventrally high metapterygoid-hyomandibular complex, and a large, rounded and convex operculum.

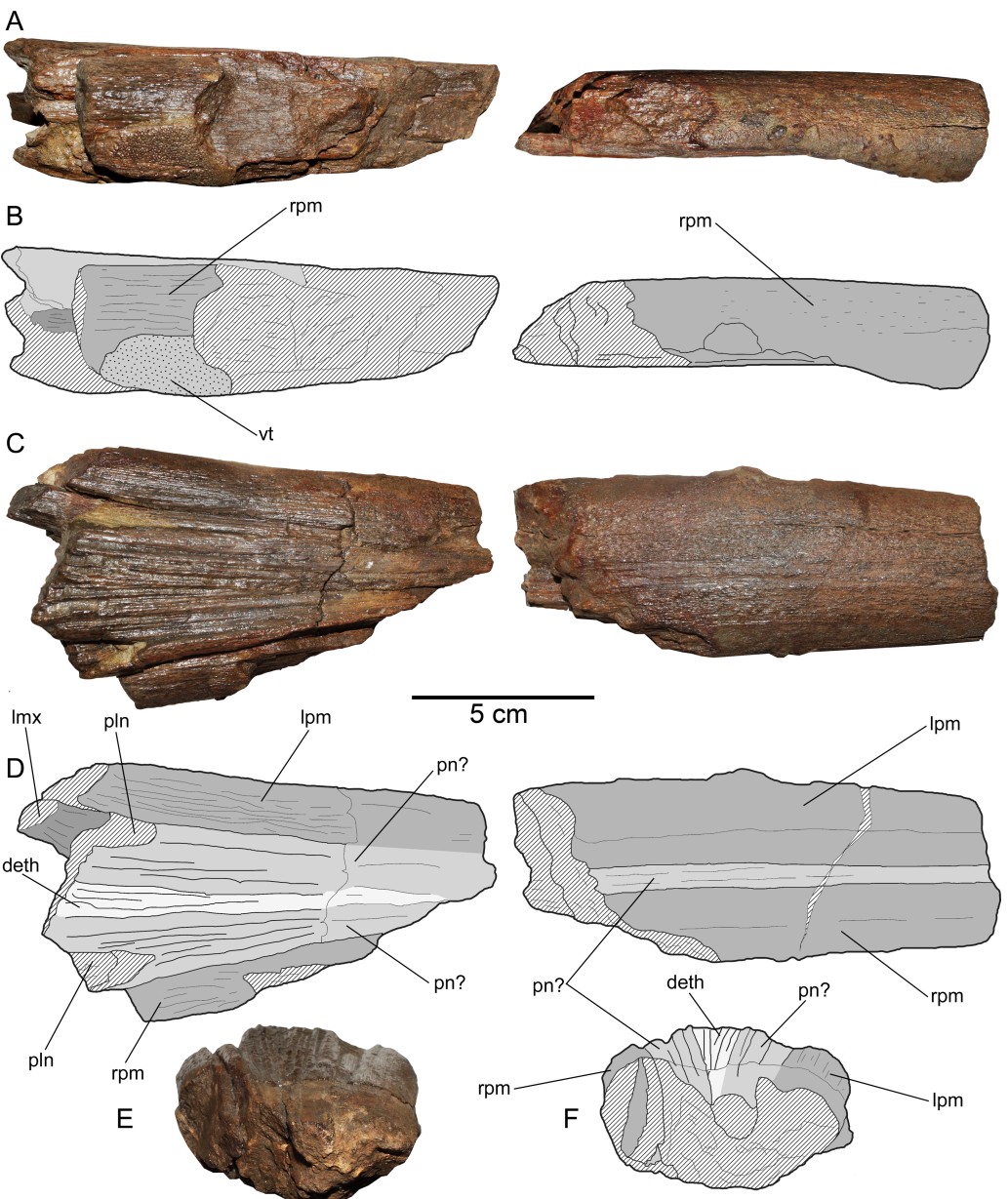

**Figure 4** ***Loancorhynchus catrillancai* gen. et sp. nov. SGO.PV.6634, holotype. Anatomy of the rostrum.** (A) Right view. (B) Line drawing interpretation of the anatomical elements. (C) same elements in dorsal view. (D) Line drawing interpretation of the anatomical elements. (E) anterior view of the anteriormost rostral fragment. (F) Scheme of the previous. Anatomical abbreviations: deth, dermethmoid; lmx, left maxillary; lpm, left premaxillary; pln, posterolateral notch; pn?, prenasals(?); rpm, right premaxillary; vt, villiform teeth.

On the other hand, the presence of dorsal rostral ridges and sulci seems to be a symplesiomorphic trait shared with *Blochius* spp. These characters support *Loancorhynchus catrillancai* within the node including *Xiphias* + *Blochius* spp.; it is also segregated from other lineages with dorsal ridges (but having more basal features), such as *Aglyptorhynchus*

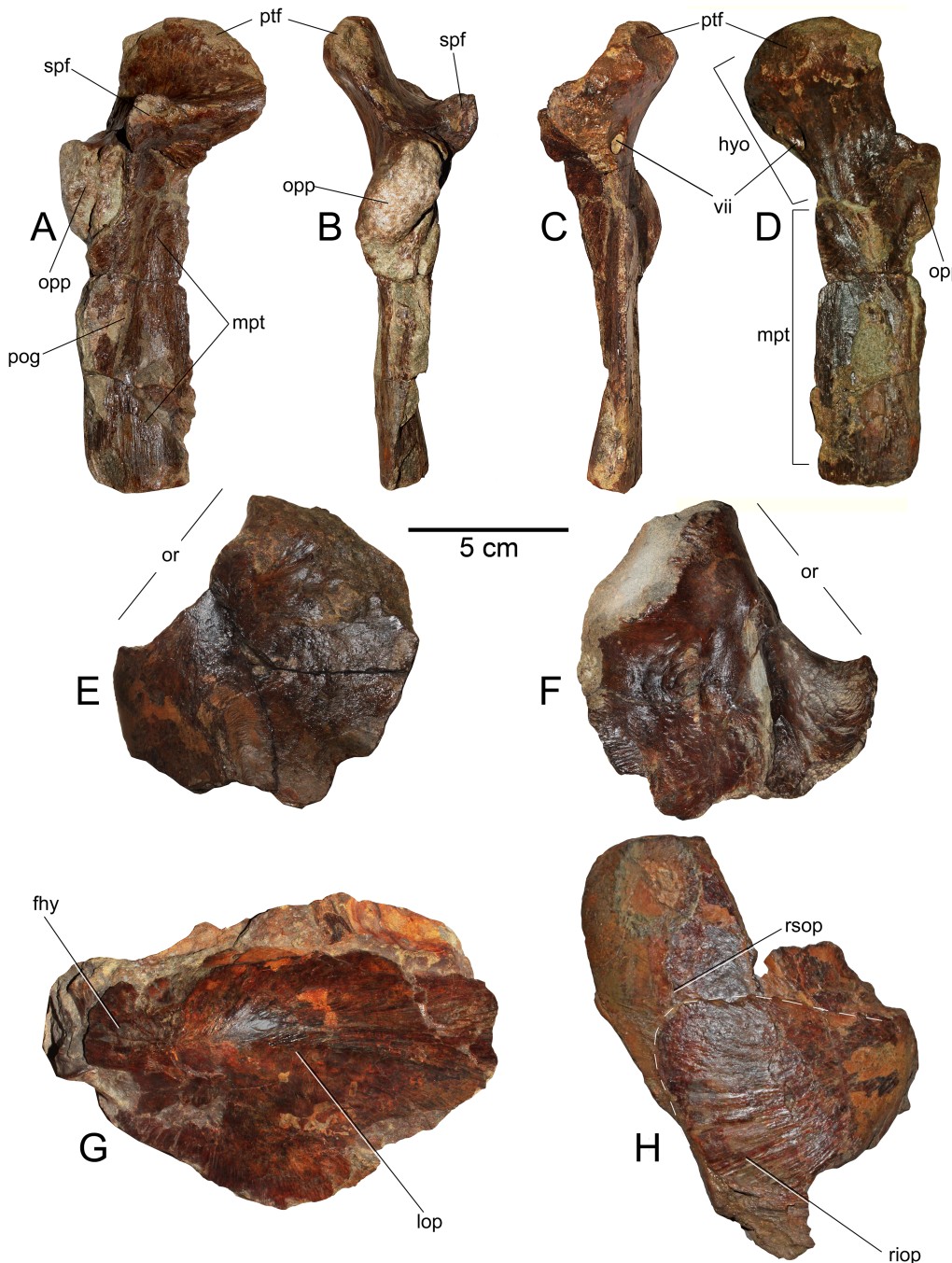

**Figure 5** *Loancorhynchus catrillancai* **gen. et sp. nov. SGO.PV.6634, holotype. Skull elements.** (A) Right hyomandibular/metapterygoid.in lateral (external) view. (B) Posterior view. (C) Anterior view. (D) Internal view. (E) Left frontal in dorsal view. (F) Same in ventral view. (G) Right opercle and supraopercular in right lateral view. (H) Left opercle in left lateral view. Anatomical abbreviations: fhy, facet for the hyomandibular; hyo, hyomandibular; lop, left opercle; mpt, metapterygoid; opp, opercular process; or, orbit; pog, preopercular groove; ptf, pterotic facet; riop, right interoperculum; rsop, right supraopercular; spf, sphenotic facet; vii, foramen for the *truncus hyoideomandibularis* of the facial nerve (VII).

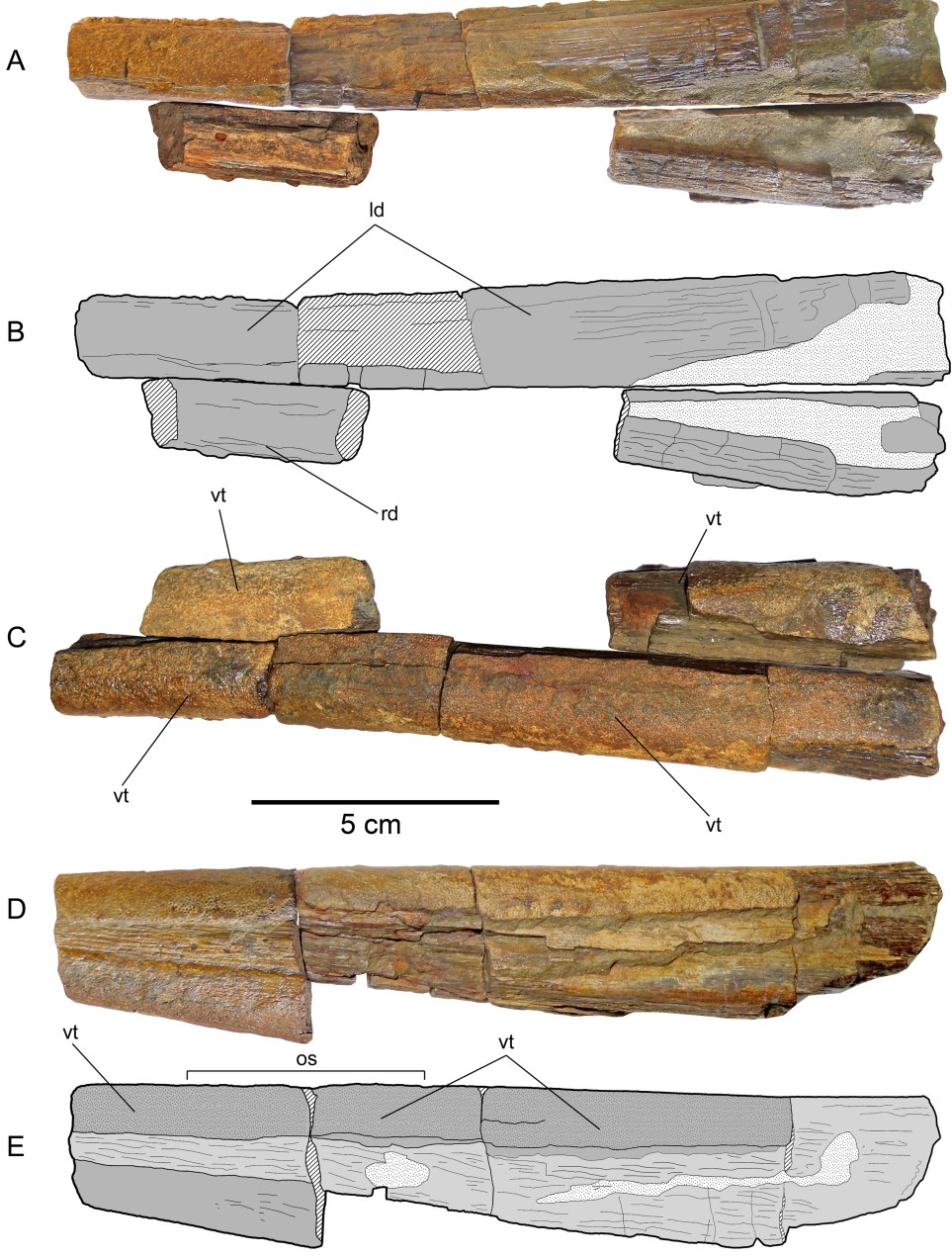

**Figure 6** *Loancorhynchus catrillancai* **gen. et sp. nov. SGO.PV.6634, holotype. Dentaries.** (A) Dentaries in ventral view. (B) Scheme of the previous. (C) Dentaries in occlusal view. (D) Left dentary (the most complete) in left lateral view. (E) Scheme of the previous. Anatomical abbreviations: ld, left dentary; os, occlusal surface; rd, right dentary; vt, villiform teeth.

and *Xiphiorhynchus*. Finally, the presence of villiform teeth over the occlusal surface of the dentaries is remarkable. Such a feature was previously described only in hemingwayids (i.e., *Hemingwaya sarissa* *Sytchevskaya & Prokofiev, 2002*). The presence of villiform teeth

in *Loancorhynchus catrillancai* dentaries could be atavic, or else, it could suggest a non-adult ontogenetic stage, as it occurs in the larval rostrum of *Xiphias* that bears villiform teeth, but these are absent in its adult stage (*Nakamura, 1985*; *Johnson, 1986*; *Fierstine & Monsch, 2002*). However, the latter hypothesis is unprobably for a large specimen such as *Loancorhynchus catrillancai*, with a length estimated in 2.7 m (see further text).

## DESCRIPTION OF *LOANCORHYNCHUS CATRILLANCAI* HOLOTYPE

**General remarks**— *Loancorhynchus catrillancai* is three-dimensionally preserved. Besides the rostrum, all the available postorbital elements are well-preserved and none of them is severely deformed. Both operculi even preserve their convexity. The rostrum is partially crushed dorsoventrally. Its anteriormost fragment is slightly recurved to the right side. On the contrary, the left dentary (the most complete) is recurved to the left side, which indicates that both deformations were caused likely by taphonomic condition instead of occurring in the living animal. During its preparation, it was evident that the skull was naturally dissarticulated and embedded in a soft, reddish, micaceous sandstone (consistent with those observed in the base of the studied section; see Geologic Setting). The length of the cranial elements in anatomic position, represents approx. 60 cm. Based on the affinities of *Loancorhynchus catrillancai* with Blochiidae, the skull could reach a conservative length of 90 cm, assuming a rostrum with a blunt tip instead a gradually reduced, sharp tip. The body length of *Loancorhynchus catrillancai* can be estimated based on the proportions of complete skeletons of *Blochius longirostris*, where the skull is ca. one third of the whole skeleton (*Fierstine & Monsch, 2002*: fig. 1). Then, *Loancorhynchus catrillancai* body length can be estimated in 2.7 m.

**Premaxillaries**—The premaxillaries form most of the anteriormost part of the rostrum.The anterior tip of the rostrum is lost; however, this was likely formed by the union of both premaxillaries, suggested by their large participation in the anteriormost available rostrum fragment. In the latter, premaxillaries are strongly fused in the mid and ventral part, being interrupted in the dorsal midline by a thin anterior extension of the prenasals(?). Both fused premaxillaries show a dorsoventrally compressed oval cross-section. Nutrient canali are not obseved, probably due to taphonomic conditions. Profuse villiform teeth cover all the ventral and lateral surface of both premaxillaries, over the whole length of the preserved rostrum (Figs. 3A and 3B). The posterior end of each premaxillary bears profuse craniocaudal striations. Two main interior canali are visible in the anterior fragment of the rostrum (Figs. 3E–3G). Smaller canali are also present, being paired with respect to the rostrum midline.

**Maxillaries**—A small fragment of the left maxillary is preserved (Fig. 4). This lies crushed under the left premaxillary. This bone is elongated with a sub-squared cross-section. Based on the available fragment, the lateral exposure of each maxillary seems to be precluded by the posterior extension of the premaxillaries.

**Prenasals(?)**—In the anteriormost available rostrum fragment, two thin bones in its dorsal midline are here interpreted as both prenasals(?) sensu *Fierstine (1990)* (Fig. 4).

In cross-section, their contact with the premaxillaries is obscured due to preservation. The posteriormost available rostrum fragment preserves the posterior part of both prenasals(?).These laterally diverge and they have a posterolateral notch (Fig. 4D), which marks the anterior margin of the orbit. Then, the anteroventral orbit is laterally conformed by the posterior extension of the premaxillary, and ventrally, by the posterior extension of the maxillary. The posterior end of both prenasals(?) have strong folds and sulci over their dorsal surface.

**Dermethmoid**—In the posterior part of the rostrum, a partially open sutural contact is visible asides the dorsal midline. This marks the contact of the prenasals(?) with the dermethmoid, which is placed in the midline and tapers anteriorly. Dorsally, the dermethmoid bears strong folds and sulci (Fig. 4). Its posterior part is missing.

**Hyomandibular**—Only the left hyomandibular is available. This element is complete and well preserved (Figs. 5A–5D). In lateral (external) view, this shows a prominent pterotic facet which is medially recurved. In the same view, the sphenotic facet is high and diverged from the pterotic facet under ca. 90°. A large opercular process is extended over the posterior margin of the hyomandibular. In posterior view, this process appears as diagonally oriented with respect to the vertical. Over the lateral (external) surface, the hyomandibular bears a dorsoventral scar, consistent with the preopercular groove. In both anterior and internal views, there is a large foramen under the pterotic facet, here interpreted as the foramen for the *truncus hyoideomandibularis* of the facial nerve (VII).

**Metapterygoid**—This bone remains strongly articulated to the hyomandibular (Figs. 5A–5D), although, its suture is still visible from the internal and lateral views. This element diagonally overlaps the hyomndibular. Laterally, it contacts with the latter through a suture adjacent to the preopercular groove. In internal view, the dorsal part of the metapterygoid is comparatively broader than its external exposure. Its ventral margin has a squared outline.

**Frontal**—The left frontal is preserved (Figs. 5E and 5F), having the medial and posterior margins missing. This element has a posterorlateral process. This indicates that the skull becomes laterally broader immediately behind the orbit. In ventral (internal) view, the frontal shows a thick ridge that broadens posteriorly.

**Operculum**—Only the left operculum is preserved (Fig. 5G) This shows the articular process for the hyomandibular. It has a well-marked convexity immediately ventral to the articulation.

**Subopercular**—Only the right subopercular is preserved (Fig. 5H). This seems to be complete. It has a dorsal rounded contour. This element remains articulated to the right interopercular.

**interopercular**—It is crushed together with the subopercular due to taphonomic conditions, but it can be distinguished as a bony element separated from the subopercular (Fig. 5H). It has a triangular outline with a convex posterior margin.

**Dentaries**—Fragments of both dentaries are preserved (Fig. 6). The left dentary is the most complete. However, its anterior part as well as its posterior end are both missing. The right dentary is represented by two separated fragments. The complete occlusal and laterodorsal surface of the left dentary are covered by profuse, small villiform teeth. Villiform teeth are also present in the two fragments of the right dentary. The left dentary

is very straight, having a slight lateral divergence in its posteriormost preserved end. Two separated fragments of the right dentary can be attached to the left dentary in anatomic position. This shows that the jaw had a large symphysis. The anterior cross-section of the preserved jaw has a sub-squared outline. The anterior part of both dentaries narrows anteriorly. This suggests that the length of the missing anterior part could be approximately 5 cm. Both mandibular rami diverge posteriorly. Based on this divergence, a temptative position of the jaw elements with respect to the rostrum can be estimated. Even if this is not accurate, it is clear that the lower jaw is much shorter than the rostrum.

**Fin rays**—Remains of least nine fin rays were recovered. All of them were included in a single block. A large incomplete fragment has an articular head, recurved with respect to the shaft. It also has a small bulk in its anterior margin. These features are present in the first pectoral ray of *Xiphias gladius*, reason by why it is identified as the latter element.

## DISCUSSION

**Historical background in billfish phylogeny**—The fossil billfish record was extensively reviewed by *Fierstine (1972)*, recognizing five families (Blochiidae, Istiophoridae, Palaeorhynchidae, Xiphiidae and Xiphiorhynchidae). To these, *Schultz (1987)* added a sixth family, Tetrapturidae, including the genera *Hemirhabdorhynchus*, *Aglyptorhynchus*, *Pseudoistiophorus*, and *Tetrapturus*. The latter author also considered the genus *Acestrus* (an istiophorid in *Fierstine, 1972*) as *incertae sedis*. Later, *Fierstine & Monsch (2002)* recovered the five families of *Fierstine (1972)*, but considered the genera *Aglyptorhynchus*, *Cylindracanthus* and *Hemirhabdorhynchus* as unsolved. The taxonomic status of *Cylindracanthus* was discussed by *Parris & Grandstaff (2001)* who suggested affinities with Acipenseriformes instead to Xiphioids. *Fierstine (2006)* reviewed again the fossil record of billfishes, considering a large set of different specimens. This author listed the families Blochiidae, Palaeorhynchidae, Xiphiidae and Istiophoridae, and the family Hemingwayidae (*Sytchevskaya & Prokofiev, 2002*). In addition, the former family 'Xiphiorhynchidae' (*sensu Fierstine, 1972*) was re-ranked to subfamily level (Xiphiorhynchinae), and included within the Xiphiidae. After, *Fierstine & Weems (2009)* distinguished *Aglyptorhynchus* from all other Palaeorhynchidae, erecting a new subfamily, Aglyptorhinchinae.

**Phylogenetic analysis**—The datamatrix of *Fierstine & Monsch (2002)* with the modifications introduced here, was first tested with Traditional Search (Wagner; 1,000 replicates; 1,000 trees to save per replication), recovering nine most parsimonious cladograms. *Acanthocybium solandri* was selected as the outgroup. Strict consensus cladogram returned a polytomy between *Loancorhynchus catrillancai*, *Xiphias gladius* and *Blochius* spp. After, IterPcr script (*Pol & Escapa, 2009*) was used for identifying '*Blochius*' *moorheadi* and *Xiphiorhynchus* as unstable taxa. Thus, '*Blochius*' *moorheadi* was subsequently pruned, while *Xiphiorhynchus* was kept because it phylogenetic placement is relevant for assessing any eventual relationship to *Loancorhynchus catrillancai*. As a result, *Xiphias gladius* was obtained as the sister taxon of a polytomic group conformed by *Blochius* spp. + *Loancorhynchus catrillancai*. Further pruning of non-xiphioid taxa *Cylindracanthus* and *Hemirhabdorhynchus*, returned the same relationships betweeen *Xiphias* and *Blochius*

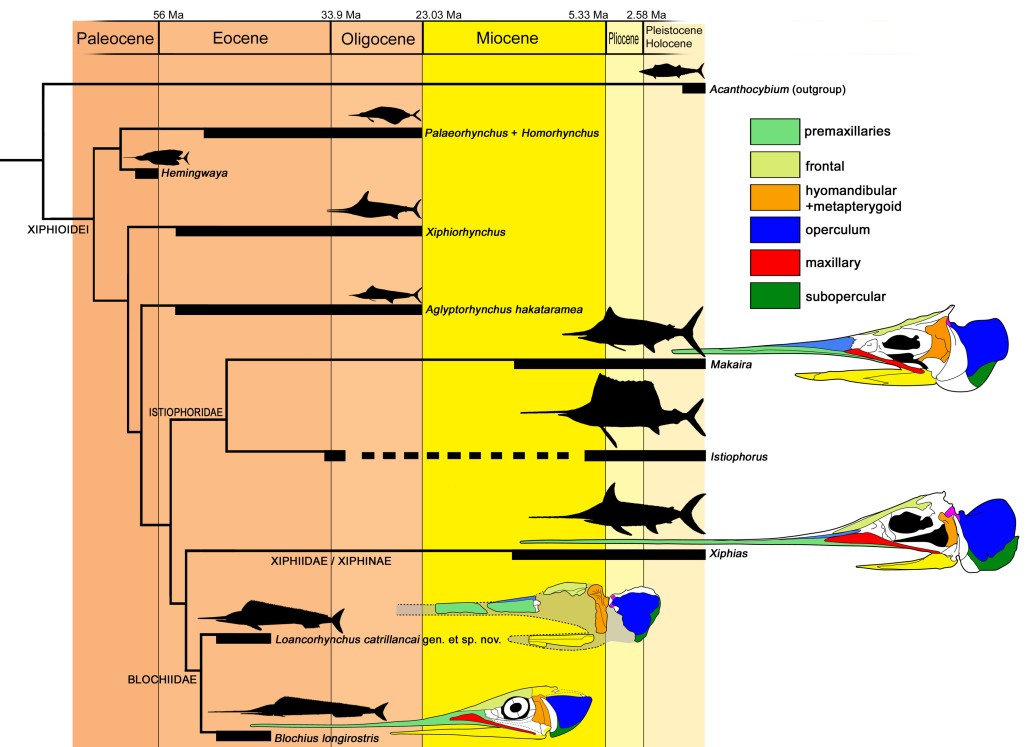

**Figure 7** **Cladogram of the Xiphioidei, including *Loancorhynchus catrillancai* gen. et sp. nov.** Clado-
gram is based on the phylogenetic analysis performed here (Single MPC; CI = 0.750; RI = 0.600; Implied
Weighting (K− = 3); New Technology Search; Ratchet. Pruned taxa: 'Blochius' moorheadi, Cylindracan-
thus, Hemirhabdorhynchus and Pseudotetrapturus luteus). Biochrons are based on *Fierstine (2006)*. Body
outlines are based in *Nakamura (1985)*. Skull schemes based in *Gregory & Conrad (1937)* and *Fierstine &
Monsch (2002)*. Dorsal longitudinal ridges on the rostrum (Char. 7) supports *Loancorhynchus catrillancai*
within the Blochidae. Villiform teeth in the rostrum and dentary (Char. 27). separates *L. catrillancai* from
*Blochius longirostris* and *Blochius macropterus*.

spp. + *Loancorhynchus catrillancai*, the latter in polytomy. *Pseudotetrapturus* was later
pruned from the previous datamatrix version, and tested with Implied Weighting, obtaining
*Loancorhynchus catrillancai* as the sister taxon of *Blochius.* As obtained synapomporphies,
the presence of dorsal longitudinal ridges on the rostrum (Char. 7) groups *Loancorhynchus
catrillancai* with the Blochiidae, while the presence of villiform teeth in the rostrum and
dentary (Char. 27) segregates *Loancorhynchus catrillancai* from *Blochius longirostris* and
*Blochius macropterus.* The topology of this cladogram is shown in Fig. 7.

In addition, the phylogenetic analysis returned *Xiphiorhynchus* as a basal lineage with
respect to *Xiphias.* Previous to this research, *Xiphiorhynchus* was obtained as the sister taxon
of *Xiphias* (*Fierstine & Monsch, 2002*). Another novel result is the obtention of *Hemingwaya*
as the sister taxon of *Palaeorhynchus + Homorhynchus.*

**Paleobiogeography**—Previous to this research, the known records of Blochiidae were
restricted to the Middle Eocene of Monte Bolca, Italy. *Loancorhynchus catrillancai* represents
the first occurrence of a blochiid in the southern hemisphere, and the fourth record of
a Paleogene xiphioid in the Weddellian Province, adding to cf. *Xiphiorhynchus* from

the Eocene of Marambio (=Seymour) Island, Antarctica (*Cione, Reguero & Elliot, 2001*), *Aglyptorynchus hakataramea*, from the late Oligocene of New Zealand (*Gottfried, Fordyce & Rust, 2013*), and the still undescribed GS 13,924 skull from the Eocene of Otago, New Zealand (*Campbell et al., 2013*). This sparse record reveals a diversity of Paleogene xiphioids in the southern hemisphere, restricted to the families Xiphiorhynchidae, Palaeorhynchidae, and now Blochiidae.

## CONCLUSIONS

*Loancorhynchus catrillancai* represents the unique Middle Eocene xiphioid known to date in the southeastern Pacific. Its first available elements (rostrum and dentaries) were initially considered as an indeterminate xiphiorhynchine (*Friedman & Otero, 2009*), while its stratigraphic provenance remained dubious since 1935, when this specimen was recovered. New preparation revealed basal features found among blochiids, xiphiorhynchines and hemingwayids. Also, derived traits exclusively known in Neogene xiphiines were found in SGO.PV.6634. Based on these facts, this research reassesses the taxonomical determination of SGO.PV.6634, being now identified as a new genus and species, *Loancorhynchus catrillancai*. This new genus and species represents the first record of the clade Blochiidae in the southern hemisphere. The new occurrence of a Middle Eocene swordfish now in the southeastern Pacific, helps to fill the Eocene-Middle Miocene gap in the xiphioid fossil record. It also represents the fourth record of a Paleogene billfish in the Southern Hemisphere, showing that this group already reached a wide distribution along the Southern Hemisphere previous to the Neogene, with Eocene records known in New Zealand (*Gottfried, Fordyce & Rust, 2013*; *Campbell et al., 2013*), Antarctica (*Cione, Reguero & Elliot, 2001*) and the new record here presented, now in South America.

### Institutional abbreviations

| | |
|---|---|
| **GS** | Geological Survey, GNS Science, Lower Hutt, Wellington, New Zealand |
| **SGO.PI** | Paleontología de Invertebrados, Museo Nacional de Historia Natural, Santiago, Chile |
| **SGO.PV** | Paleontología de Vertebrados, Museo Nacional de Historia Natural, Santiago, Chile |

## ACKNOWLEDGEMENTS

Special thanks are due to H Fierstine (California Polytechnic State University) who provided valuable literature during early stages of this research. Thanks also to M Friedman (University of Oxford) for his contribution in the preliminary study of SGO.PV.6634. AM Prokofiev (AN Severtsov's Institute of Ecology and Evolution, Russian Academy of Sciences) is also thanked by valuable literature provided. K Monsch (Naturalis Biodiversity Center, Netherlands) is acknowledged for the revision of a early version of the manuscript. M Gottfried (Geological Sciences and Museum, Michigan State University, USA), P Brito (Universidade do Estado do Rio de Janeiro, Brasil), D Parris (New Jersey State Museum, USA) and G Piñeiro (Universidad de la República, Uruguay) are especially thanked

for their reviews and all the comments that improved this paper. RE Fordyce (Otago University, Dunedin, New Zealand) is acknowledged for allowing access to the holotype of *Aglyptorhynchus hakataramea* during 2013.

### Funding
This research was supported by Proyecto Anillo ACT-172099 (Conicyt-Chile). The funders had no role in study design, data collection and analysis, decision to publish, or preparation of the manuscript.

### Grant Disclosures
The following grant information was disclosed by the author:
Proyecto Anillo ACT-172099 (Conicyt-Chile).

### Competing Interests
Rodrigo A. Otero is a business partner and co-founder of Paleosuchus Ltda. Consultory.

### Author Contributions
- Rodrigo A. Otero conceived and designed the experiments, performed the experiments, analyzed the data, contributed reagents/materials/analysis tools, prepared figures and/or tables, authored or reviewed drafts of the paper, approved the final draft.

### Data Availability
Specimens are available at the Área Paleontología, Museo Nacional de Historia Natural: SGO.PV.6634.

### New Species Registration
The following information was supplied regarding the registration of a newly described species:
Publication LSID: urn:lsid:zoobank.org:pub:ED22325A-7C78-469A-9985-365C31064652
*Loancorhynchus* name LSID: urn:lsid:zoobank.org:act:FA566753-31D4-45B5-9E7A-03F8C7526E7D
*Loancorhynchus catrillancai* gen. et sp. nov. LSID: urn:lsid:zoobank.org:act:8FFA9EE9-A81B-4468-9FBA-9D6C306D86C8.

### Supplemental Information
Supplemental information for this article can be found online at http://dx.doi.org/10.7717/peerj.6671#supplemental-information.

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
