# Peer review of "Loancorhynchus catrillancai gen. et sp. nov., a new swordfish (Xiphioidei, Blochiidae) from the Middle Eocene of central Chile"

_PeerJ, doi:10.7717/peerj.6671_

## Round 0.1 · original submission · Minor Revisions

Dear Dr. Otero,

We have now three review reports for your article on “Loancorhynchus catrillancai gen. et sp. nov., a new swordfish (Xiphioidei, Blochidae) from the middle Eocene of central Chile”. I am glad to say that all the reviewers found the manuscript as a very interesting contribution to the knowledge of the fossil swordfishes by describing a new species. In that context, the reviewers found just minor corrections that should be addressed before to an eventual acceptation of the manuscript for publication in PeerJ. Please, pay attention to the References section where the reviewers have found some inconsistencies and if possible, try to ask help for a native English speaker to make an improvement of the language.

I very appreciate that you have agreed to make some changes to the Systematic section, particularly on the derivation of the new species name. Many thanks.

I hope you find useful the review reports and can address your manuscript accordingly.- If so, I shall expect to receive the revised version of your article very soon.

Best regards,
Graciela Piñeiro

·

Basic reporting

The ms. is generally readable and readily understood, but there are some awkwardly worded passages and misspelled words reflecting the fact that English is a second language for the author. I recommend that the author have someone completely fluent in English take one more editorial pass through the ms.

For ease of use by readers, I recommend that the author spell out and explain abbreviations for institutional collections when they are first used in the text (e.g. ‘GS 13924’ on p. 7 of the ms. pdf) rather than just listing them in one place.

Family name should be BLOCHIIDAE, not BLOCHIDAE as it is spelled on ms. p. 14. Likewise XIPHIAS not XIPHIUS (p. 6).

The author should define and cite a source or sources to better explain the Weddellian Biogeographic Province.

I don’t understand what ‘strongly attached … because of taphonomic reasons’ means, in the description of the opercular series on p. 19.

The Taxonomic Placement section that is in the Discussion should be in the Systematic Paleontology section, it is much more description than discussion.

I found Fig. 1A a bit difficult to interpret – is the coastline of Chile indicated on this map on the left edge of the gray-shaded area? If so it would be helpful to just add ‘Pacific Ocean’ to the map to make that clear.

On Fig. 2, the abbreviation dr is not explained in the caption, and the abbreviation fr is not in bold as are the other abbreviations in the caption. Also, what is meant by ‘anterior bulk?’

On Fig. 4, I suggest using the wording ‘line drawing interpretation of element’ rather than the awkward phrase ‘scheme of the previous.’

On Fig. 7, it would be helpful to see the supporting characters (by number) mapped on, or at least an explanation of where in the paper that information is provided.

Experimental design

The scope of comparisons in the Diagnosis needs to be expanded – in other parts of the paper the author makes comparisons between the new taxon and several previously described billfish taxa including Aglyptorhynchus and Hemingwaya, but then only makes comparisons with Blochius, Xiphias and Xiphiorhynchus in the Diagnosis itself.

The author needs to include a clear statement to justify and support his placing the new taxon in the Family Blochiidea as he makes several comparisons between his new taxon and other genera that are assigned to a variety of billfish families. I found it difficult to follow his train of thought (e.g. see pp. 15 and 16, and then p. 22) as he goes from the individual features he describes on the new taxon, to the comparisons with other billfish taxa that each share a different subset of those features, to his ultimate taxonomic decision on where to assign his new taxon. The phylogenetic analysis does not help as the new taxon is part of an unresolved polytomy in the strict consensus cladogram.

The statement on p. 16 that the postorbital skull elements are well-preserved and undistorted is overstated and inconsistent with the actual details of the Description, which notes incomplete and somewhat crushed/distorted, and/or fragmentary, elements, and well as the fact that many skull elements are simply not represented at all.

Validity of the findings

I am very skeptical that a billfish estimated to be nearly three meters in length is exhibiting a LARVAL condition with respect to the presence of villiform teeth, as the author suggests on p. 22.

As noted in other sections of the review, I believe a clearer and more comprehensive case can be made for the taxonomic decisions arrived at, with respect to the phylogenetic conclusions, and a more comprehensive Diagnosis should be provided.

Additional comments

Overall this is a well-executed paper describing a noteworthy specimen that should be published. The anatomical descriptions are generally solid and the figures are clear and at a publishable standard. I do believe that the paper needs one more editorial pass by someone completely fluent in English to edit out some awkward wording and misspellings, and as I noted I am still a bit confused on the rationale for the family-level assignment and how that assignment relates to the phylogenetic analysis that is presented, and I believe the Diagnosis should be strengthened by including all mentioned and relevant taxa.

·

Basic reporting

This is an excellent article dealing with the description of a new genus and a new species of a Blochidae xiphioid. The article is clear, with interesting results, update references (but see below) and excellent figures.

However, I would like to suggest that:
1- References should be checked;
2 - After the description of the new taxon in the "Systematic Palaeontology" (line 218), the specimen ceased to be treated by the number SGO.PV.6634 and started being treated by the generic or (better) the specific name Loancorhynchus catrillancai.

Experimental design

No comments.

Validity of the findings

Nothing to add.

Additional comments

I would suggest to the author to check the references because I did not find some in the text (cf. Cappetta 1987, Chirino-Galvez 1993, Fierstine 2oo7, Fierstine and Starnes 2005, Kummel 1964, Muizon and DeVries, 1985....) or others in the references (Momsch and Fierstine, 2002).

Also, that after the first description of the new taxon, the specimen ceased to be treated by the number SGO.PV.6634 and started being treated by the generic or (better) the specific name Loancorhynchus catrillancai.

There are still some minor typos throughout the text.

Finally, I would like to congratulate the author for this work.

·

Basic reporting

The article meets the editorial standards in my opinion. It is properly organized in standard format for a paleontological descriptive work, including the systematic portion. I agree that it qualifies as a new genus and species; the description meets prevailing standards. The bibliography is generally sufficient, but requires some relatively minor revisions, as noted in my attached word file. That is, in fact, where most revisions are needed. I have also noted other typographical and style comments, notably for standard English expressions. If the editors wish me to do so, I will be pleased to review the article again following revision.

REVIEW OF: LOANCORHYNCHUS CATRILLANCAI MANUSCRIPT
TYPOGRAPHICAL AND STYLISTIC NOTES:
Line 18: these being (conventional English sequence)
Line 22: Omit the word “studied”
Line 23: such a unique (conventional expression)
Line 24: Omit the word “studied”
Line 28: Omit the word “studied”
Line 71: crop out (conventional expression)
Line 79: Is Sernageomin, 2003 in the bibliography?
Line 114: Spell out generic name.
Line 114: phragmocone (usual English spelling)
Line 133: collection (not recollection)
Line 159: spelling of Schultz?
Line 169: What is the date of the Nakamura reference 1985? 1995?
Line 173: Should this be Fierstine and Monsch?
Line 181: spelling of Schultz?
Line 263: Abbreviation should be “approx.”
Line 265: based on the (conventional expression)
Line 267: Should this be Fierstine and Monsch?
Line 274: Prefer “nutrient” to “nutritious”.
Line 302: there (spelling)
Line 310: “the”, rather than “their”
Line 355: Is Fierstine, 1972 in the bibliography? (also Lines 347, 348, 355.)
Line 378: Should this be Fierstine and Monsch? (also Lines 382 and 394.)
Line 394: Check Nakamura publication date (as for Line 169).
Line 401: Omit word “shows”.
Line 419: Was Paleogene intended (instead of Paleocene)
Line 421: Should this be Gottfried et al.?
Line 425: Special thanks are due to H. Fierstine (conventional expression)
Line 429: for valuable literature (conventional expression)
Line 449: was this cited?
Line 458: Was this cited?
Line 459: Kent, Ohio, U.S.A.
Line 458: Was this cited?
Line 473: Capitalize Scombroidei, Middle, Eocene, Monte, Miscellanea
Line 476: Was this cited?
Line 506: Was this cited?
Line 519: This is not entered in alphabetical order.
Line 533: Check date of reference.
Line 541: Was this cited?
Line 550: Was this cited?Line 560: Was this cited?
Line 563: Enter entire reference.
Line 577: Was this cited?
Line 599: Should this be Fierstine and Monsch?
Figure 2 Caption. Should this be Fierstine and Monsch?

Experimental design

As noted previously, the organization of the paper is standard for descriptive and systematic works. The description is very thorough.

Validity of the findings

I agree with the conclusions, that is, that it is a new genus and the corollary conclusions are valid.

Additional comments

I have no further comments.

---

## Round 0.2 · Minor Revisions

Dear Dr. Otero,

Thank you very much for to have submitted the revised version of your manuscript “Loancorhynchus catrillancai gen. et sp. nov., a new swordfish (Xiphioidei, Blochidae) from the middle Eocene of central Chile”. Indeed, it is a nice article and after a careful reading, I am almost sure that it will be acceptable for publication in PeerJ. I found some typos that should be fixed and also some problems with the presence of the maxilla, and some missing references in the text and in the References list that will be of very easy and quickly solution. Thus, please, see the annotated PDF file that I am attaching to this letter, make the requested modifications and resubmit your article. Many thanks.

With my best wishes,
Graciela Piñeiro

---

## Round 0.3 · Minor Revisions

Dear Dr. Otero,

Thank you for submitting the revised version of your interesting manuscript about the description of a new species of swordfish from the Middle Eocene of Chile.
I am very sorry for my mistake about the posterolateral notch; I indeed did not see it in figure 4.
Well, I have marked just a few more changes to do before acceptation of your manuscript for publication in PeerJ, they will be very easy to fix (see the annotated PDF attached).
Kind regards,
Graciela Piñeiro

---

## Round 0.4 · accepted · Accept

Dear Dr. Otero,

I have read the manuscript once again and I do not have concerns about your corrections in this last version of your manuscript as they do not modify previous statements. Thus, I am glad to announce that your article is ready for to be published in PeerJ. Congratulations!

With my best regards,

Graciela Piñeiro

#